# Local Climate Adaptation in Chinese Indigenous Pig Genomes

**DOI:** 10.3390/ani15162412

**Published:** 2025-08-18

**Authors:** Yuqiang Liu, Yang Xu, Guangzhen Li, Wondossen Ayalew, Zhanming Zhong, Zhe Zhang

**Affiliations:** State Key Laboratory of Swine and Poultry Breeding Industry, Guangdong Provincial Key Lab of Agro-Animal Genomics and Molecular Breeding, College of Animal Science, South China Agricultural University, Guangzhou 510642, China; yuqiangliu123@126.com (Y.L.); yangxu202309@163.com (Y.X.); guangzhenli6228@126.com (G.L.); wondessenayalew9@gmail.com (W.A.); zhongzhanming204@163.com (Z.Z.)

**Keywords:** local adaptation, genotype–environment association (GEA), Chinese indigenous pigs

## Abstract

Chinese indigenous pigs have evolved in a wide range of climates, making them an excellent model for studying how animals adapt to local environments. In this study, we analyzed whole-genome data from 578 pigs representing 46 native breeds across China and linked genetic differences to environmental conditions, especially precipitation during the wettest season. We found that precipitation plays a key role in shaping the genetic makeup of these pigs. Many of the identified genes are involved in immune function and metabolism, and one gene, MS4A7, stood out as a strong candidate for adaptation to precipitation patterns. Our findings help explain how native pigs have adapted to their environments and provide valuable information for conserving genetic diversity and developing climate-resilient breeding programs.

## 1. Introduction

The domestic pig (*Sus scrofa domesticus*) is among the most economically and scientifically valuable livestock species globally. As a primary meat source, pigs contribute significantly to global food security and rural economies. Beyond their agricultural importance, pigs also serve as essential biomedical models, owing to their physiological and anatomical similarities to humans. They are widely used in studies of xenotransplantation, metabolic disorders, and immunology [1,2].

Recent advances in high-throughput genotyping and sequencing technologies have greatly accelerated pig population genomics research. These efforts have uncovered complex patterns of domestication, introgression, and local adaptation in both commercial and indigenous pig populations [3,4,5]. Notably, native Chinese pig breeds display exceptional genetic diversity and phenotypic variation, shaped by geographic isolation, distinct breeding practices, and ecological adaptation [6,7,8]. Understanding their local adaptation is crucial for conserving genetic resources and optimizing breeding strategies under changing environments.

Gene–environment association (GEA) analysis is a powerful approach for identifying genomic loci involved in local adaptation through correlations between allele frequencies and environmental variables [9,10]. GEA methods—such as latent factor mixed models (LFMMs), redundancy analysis (RDA), and gradient forest—have been widely used in plant species like *Arabidopsis thaliana* [11] and maize [12], and increasingly in animals such as fish, birds, and domestic ruminants [13,14,15,16]. However, applications of GEA in pigs remain limited, and few studies have integrated genomic, environmental, and regulatory layers to elucidate the adaptive landscape of indigenous breeds. Among climatic variables, BIO16 (precipitation of the wettest quarter) represents a critical ecological factor, particularly in subtropical and monsoon-influenced regions like southern China. It reflects seasonal water availability, which can modulate host–pathogen dynamics, thermoregulation, and hydric stress tolerance [11]. In humid and pathogen-rich environments, increased precipitation may impose selection on immune pathways, mucosal barrier function, and metabolic regulation [10]. Studies in birds and domestic ruminants have reported strong associations between precipitation variables and adaptive divergence in immunity-related genomic regions [16,17]. Therefore, focusing on BIO16 in the context of Chinese indigenous pigs may reveal key mechanisms of precipitation-driven local adaptation, with implications for climate-resilient livestock breeding strategies. In this study, we investigate the genomic basis of local environmental adaptation in 46 Chinese indigenous pig breeds through an integrative framework combining population genomics, genotype–environment associations, and functional annotations. We identify key environmental drivers of genomic divergence and assess potential adaptive responses under future climate scenarios, providing critical insights for conservation and climate-resilient breeding.

## 2. Materials and Methods

### 2.1. Sample Collection and Genotypic Data Processing

Whole-genome resequencing data were retrieved from the Pig Genome Resequencing Project (PGRP v1) [18]. To minimize the confounding effects of recent selection and admixture, commercial and hybrid breeds were excluded. Only indigenous Chinese breeds with clearly documented geographic origins were retained. To ensure breed assignment consistency and eliminate genetically redundant individuals, pairwise identity-by-state (IBS) analysis was conducted using PLINK v1.90 [19]. Individuals exhibiting IBS values greater than 0.90 were excluded from the dataset. After quality control, the final dataset comprised 578 individuals representing 46 indigenous pig breeds (Appendix A). SNPs with a minor allele frequency (MAF) < 0.1 and a missing rate > 0.1 were filtered using PLINK v1.90 [20]. MAF filtering was applied at the whole-dataset level (rather than within individual breeds) to remove ultra-rare variants and enhance the robustness of downstream analyses.

### 2.2. Environmental Data Collection and Preprocessing

Breed-level geographic coordinates were determined based on historical records and documented breed distribution data. A single representative latitude–longitude coordinate was selected for each breed to reflect its long-term local adaptation. Nineteen bioclimatic variables (BIO1–BIO19) (Appendix A) were extracted from the WorldClim v2.1 database (https://www.worldclim.org/, accessed on 14 March 2025) at a ~5 km (~2.5 arc-min) spatial resolution, representing long-term climatic averages from 1900 to 2000 [19].

### 2.3. Population Structure and Genetic Diversity Analysis

Principal component analysis (PCA) and pairwise identity-by-state (IBS) matrices were calculated using PLINK v1.90 [20]. A neighbor-joining (NJ) tree was built using MEGA v11 [21] based on the genetic distance matrix and visualized using the R package ggtree V3.16.3 [22]. To reduce LD redundancy, SNPs were pruned using PLINK (--indep-pairwise 500 50 0.1), resulting in a final set of 226,394 SNPs. ADMIXTURE v1.3.0 [23] was employed to estimate ancestry proportions using K = 2–10. Based on clustering patterns and geographic origin, breeds were categorized into four regional groups: ECN (East Central), SCN (South Central), SWCN (Southwest), and NCN (North Central). VCFtools v0.1.17 [24] was used to calculate nucleotide diversity (π) and Tajima’s D across each group in non-overlapping 50-kb windows (--window-pi 50000; --TajimaD 50000).

### 2.4. Gene–Environment Association Analysis

We performed genome-wide GEA using the latent factor mixed model (LFMM) implemented in the R package LEA [25,26]. To adjust for population structure, we selected the number of latent factors (K) based on PCA and ADMIXTURE results. Specifically, PCA revealed genetic differentiation among geographic groups, and ADMIXTURE was conducted for K = 2–10. Considering the genetic structure, geographic distribution, and domestication history of Chinese indigenous pigs, K = 4 was chosen for subsequent LFMM analysis. The analysis was based on genotypes from 578 individuals representing 46 indigenous pig breeds in China. Nineteen bioclimatic variables (BIO1–BIO19) obtained from the WorldClim v2.1 database were used as environmental predictors. SNPs with *p* < 1 × 10^−5^ were considered to be suggestively associated, following the original LFMM framework, where this threshold corresponds approximately to |z| > 4, offering a conservative control for false positives in genome-wide scans [27].

### 2.5. Environmental Variable Selection

PCA of 19 bioclimatic variables was performed using the R packages FactoMineR and factoextra [28]. To address multicollinearity, pairwise Pearson correlation coefficients were computed and visualized with the corrplot package. Environmental variable importance in explaining genetic variation was assessed using the gradientForest R package [29]. SNPs (LD-pruned with\--indep-pairwise 500 50 0.1) were used to calculate population-level allele frequencies per breed [29,30]. The gradient forest model was fitted with 500 trees, and variable importance was evaluated based on both split accuracy and cumulative R^2^. To reduce multicollinearity and retain the most informative predictors for downstream analyses, we first computed a Pearson correlation matrix among the 19 bioclimatic variables and excluded one variable from any pair with |r| ≥ 0.8. We then ranked the remaining variables by importance using the gradientForest model (500 trees; importance summarized by split accuracy and cumulative R^2^) based on the joint criteria of low pairwise correlation (|r| < 0.8) and high gradientForest importance. For RDA, we used SNPs that were pruned for LD across all 578 individuals. Model significance and the significance of each constrained axis were assessed using a permutation test with 999 permutations, and the percentage of variance explained by each axis was calculated.

### 2.6. Environmental vs. Geographic Contributions to Genetic Differentiation

We employed Mantel and partial Mantel tests to evaluate the relative contributions of geographic isolation-by-distance (IBD) and environmental isolation-by-environment (IBE) to genetic differentiation. Pairwise F*_ST_* values were calculated separately for neutral variants (LD-pruned SNPs) and putatively adaptive variants (LFMM-associated SNPs, *p* < 1 × 10^−5^). Geographic distances were derived from great-circle distances between breed-level coordinates, while environmental distances were calculated as Euclidean distances using standardized values of all 19 bioclimatic variables. Statistical significance was assessed through 999 permutations using the mantel() and mantel.partial() functions in the R package vegan.

### 2.7. Functional Annotation and Enrichment Analyses

Functional annotation was performed for the SNPs significantly associated with BIO16 (*p* < 1 × 10^−5^): SnpEff v4.3 [31] for genomic feature classification, chromatin state annotations from 14 porcine tissues [32] to define regulatory elements (e.g., promoters, enhancers), and gene expression profiles from the PigGTEx database [18] to identify tissue-specific transcriptional activity. Tissue specificity was quantified using t-statistics as described by Finucane et al. [33], and the top 1000 genes with the highest t-values in each tissue were selected as tissue-specific gene sets. Enrichment of significant SNPs within these sets was assessed using Fisher’s exact test. Enrichment ratios were calculated using the oddsratio() function in the R package fmsb [34], and statistical significance was tested using 10,000 permutation replicates implemented in regioneR. Details of the analytical framework are available in reference [35]. To identify candidate genes, we defined a 50 kb window centered on each significant SNP (±25 kb). Gene coordinates were retrieved from the Sus scrofa 11.1 genome annotation file (GTF format), and intersections between SNP windows and gene bodies were determined using BEDTools v2.25.0 [36]. Gene Ontology (GO) and Kyoto Encyclopedia of Genes and Genomes (KEGG) pathway enrichment analyses were conducted using the clusterProfiler R package [37], applying default settings.

### 2.8. MS4A7 Locus Analysis and Selection Scan

We focused on the top-ranked SNP, which showed the strongest association with BIO16. To investigate potential selective pressure at this locus, we stratified the dataset into high (top 10%) and low (bottom 10%) BIO16 groups, with 58 individuals in each. Cross-population extended haplotype homozygosity (XP-EHH) scores were computed using selscan v2.0 [38] in non-overlapping 2 kb windows, designating the high-BIO16 group as the test population. Weir and Cockerham’s F*_ST_*, nucleotide diversity (π), and Tajima’s D were also calculated using VCFtools v0.1.17 in matching 2 kb windows. Linkage disequilibrium (LD) decay around the SNP was evaluated using PLINK v1.90, with the parameters --r2, --ld-window-kb 200, and --ld-window-r2 0. Genotype–phenotype associations were examined using Student’s *t*-tests. Chromatin state annotations were obtained for 14 representative porcine tissues, covering 14 chromatin states indicative of key regulatory features (e.g., active promoters, enhancers, repressed regions), based on ENCODE-like epigenomic resources [32]. Meanwhile, gene expression profiles of *MS4A7* across 35 tissues were retrieved from the PigGTEx database [18].

## 3. Results

### 3.1. Population Genetic Structure and Diversity

We conducted genome-wide population genetic analyses on 46 Chinese indigenous pig populations (Figure 1a, Appendix A). Genotype filtering was performed using a MAF < 0.1 and a missing rate threshold of >0.1, yielding 17,700,714 high-quality SNPs from an initial total of 33,485,897. After further LD pruning with the --indep-pairwise 500 50 0.1 option, a total of 226,394 SNPs were selected for subsequent analysis. PCA based on 226,394 high-quality SNPs revealed a clear population structure. Populations from the East China subclade (ECN) were distinctly separated along the first principal component (PC1), while populations from South China (SCN), Southwest China (SWCN), and North China (NCN) displayed further separation along PC2 (Figure 1b). ADMIXTURE analysis (K = 2 and K = 3) further supported these findings, showing a consistent genetic structure pattern with ECN populations exhibiting a distinct ancestral component at K = 3 (Figure 1c). Nucleotide diversity (π) analysis indicated that SCN (π = 0.003389) and SWCN (π = 0.003274) populations harbored higher genetic diversity levels compared to ECN populations (π = 0.002789), suggesting increased genetic polymorphism in southern populations (Figure 1d). Genome-wide assessments of Tajima’s D values showed consistently positive values across all four population groups (Figure 1e), with SCN and SWCN populations having slightly higher averages (1.47) than ECN (1.43) and NCN (1.18). These values reflect regional differences in the distribution of allele frequencies across Chinese indigenous pig populations.

### 3.2. Environmental Gradient Analysis and Variable Selection

We conducted a PCA on the distributions of 46 Chinese indigenous pig populations based on 19 bioclimatic variables. The first three principal components explained a total of 91.7% of the environmental variation, with PC1 and PC2 accounting for 64.7% and 19.3%, respectively. Populations from ECN, NCN, and SWCN exhibited compact distribution patterns in PCA space, suggesting homogeneous environmental backgrounds across these groups. In contrast, SCN populations were more widely scattered, indicating greater ecological diversity or adaptive plasticity in this region (Figure 2a). The biplot revealed distinct loadings and directional contributions of environmental variables in the principal component space. Notably, BIO4, BIO3, and BIO7 contributed strongly to PC1, whereas BIO8 and BIO15 had higher loadings on PC2. Scatterplot analysis based on environmental PCA further revealed fine-scale differentiation among pig populations (Figure 2b).

To reduce redundancy and identify key ecological drivers, we performed pairwise correlation analysis and PCA-based importance ranking of the 19 bioclimatic variables (Figure 2c). The correlation matrix showed that many variable pairs had Pearson correlation coefficients > |0.8|, indicating strong collinearity. Variables exhibiting high pairwise correlations were excluded to avoid multicollinearity in downstream analyses. We further employed gradient forest modeling with the “gradientForest” R package to evaluate and rank the importance of each variable (Figure 2d). Based on contribution scores and low inter-variable correlation, we selected six representative bioclimatic variables: BIO2, BIO3, BIO4, BIO8, BIO15, and BIO16. These variables comprehensively capture major ecological gradients in temperature and precipitation and act as informative predictors of environmental heterogeneity across Chinese indigenous pig populations.

### 3.3. Environmental Gradients Influence Pig Genomic Structure

We performed RDA based on 226,394 SNPs that were pruned for linkage disequilibrium (LD) across 578 individuals from Chinese indigenous pig populations to assess the influence of environmental variables on genomic variation. A permutation test (n = 999) confirmed that all six environmental predictors (BIO2, BIO3, BIO4, BIO8, BIO15, and BIO16) were significantly associated with genomic variation (*p* < 0.01) (Appendix A). The first three constrained axes (RDA1–RDA3) explained 44.5%, 18.8%, and 13.3% of the constrained variance, respectively, accounting for 76.6% of the total explained variation (Appendix A). For example, pigs from East China (ECN) aligned with gradients of BIO4, while those from Southwest China (SWCN) were mainly associated with precipitation indices (BIO15, BIO16). Additional variables such as BIO3 and BIO8 also exhibited notable effects (Figure 3a,b).

To disentangle the relative effects of geography and environment on genetic differentiation, we conducted Mantel and partial Mantel tests using FST matrices derived from environment-associated SNPs (n = 8644; *p* < 1 × 10^−5^) and putatively neutral SNPs (n = 226,394). Environment-associated SNPs showed a significant correlation with geographic distance (Mantel’s r = 0.46, *p* = 0.001), while neutral SNPs did not (r = 0.08, *p* = 0.178). When geographic distance was controlled, a significant correlation between FST and environmental distance remained for environment-associated SNPs (partial Mantel’s r = 0.25, *p* = 0.001), but disappeared for neutral SNPs (r = −0.13, *p* = 0.930), suggesting an independent role of environmental selection in shaping population differentiation (Figure 3c,d).

### 3.4. Functional Annotation of BIO16-Associated Loci

To assess the impact of MAF thresholds on GEA outcomes, we compared the results of LFMM analysis using two different filters: the default MAF > 0.1 and a more lenient MAF > 0.05. The topological patterns of significant SNPs remained consistent across these two thresholds, and the association signals, measured as −log10(*p*), showed an exceptionally high Pearson correlation (r = 0.992) between the two datasets (Appendix A). This strong correlation supports the robustness of our findings, indicating that our GEA results are stable across different MAF cutoffs.

To determine an appropriate window size for candidate gene identification, we calculated linkage disequilibrium (LD) decay across all pig populations using the PopLDdecay tool. Pairwise r^2^ values were computed to evaluate the distance at which LD decays significantly. As shown in Appendix A, r^2^ values declined rapidly within the first 40–50 kb and plateaued thereafter, suggesting that most LD was confined to a 50 kb window. Therefore, we selected a ±25 kb window around each significant SNP, which allows for the capture of the majority of LD-linked regions while minimizing the inclusion of distant, unrelated genes.

A total of 8644 SNPs showed suggestive associations with environmental variables (*p* < 1 × 10^−5^), including 310 SNPs associated with BIO16, a key environmental variable related to precipitation in the wettest quarter (Figure 4a). Based on prior analyses, we identified BIO16 as a major factor shaping the geographic and genomic landscape of Chinese indigenous pig populations. To explore how local pigs respond at the molecular level to selective pressures from precipitation gradients, we focused on BIO16. We then performed multilayered annotation and functional analysis of the significantly associated genomic loci to elucidate potential adaptive mechanisms. This approach enabled us to identify key loci likely involved in local adaptation to varying precipitation environments.

Genomic feature enrichment analysis revealed that these SNPs were significantly enriched in upstream regulatory regions (fold enrichment = 1.83, *p* = 2.09 × 10^−11^) and intergenic region (fold enrichment = 1.4, *p* = 3.67 × 10^−13^), indicating potential cis-regulatory effects (Figure 4b). Conversely, no significant enrichment was detected in non-functional regions such as 5′ and 3′ untranslated regions (UTRs) or synonymous coding sites. Chromatin state enrichment analysis (Figure 4c), based on 14-state chromatin annotations across 14 porcine tissues, revealed significant enrichment of SNPs in active regulatory elements—including enhancers and open chromatin—in tissues such as the cortex, jejunum, and duodenum. These enrichments suggest that BIO16-associated SNPs may influence gene regulation in a tissue-specific manner. Tissue-specific gene enrichment analysis (Figure 4d) further demonstrated that significant SNPs were overrepresented in the top 1000 most specifically expressed genes in tissues including the lung, spleen, hypothalamus, intestine, and adipose tissue. These enrichments were statistically significant (*p* < 0.05), with fold enrichment consistently above 1, highlighting their potential biological relevance. In total, 147 candidate genes were identified within ±25 kb flanking windows of significant SNPs (Appendix A). GO (Appendix A) and KEGG (Appendix A) pathway enrichment analyses were subsequently performed using the clusterProfiler package. Results revealed significant enrichment in pathways associated with immune responses and intracellular signal transduction. In particular, the “C-type lectin receptor signaling pathway,” involving genes such as *CALML4*, *PPP3CB*, and *NFKB2*, was significantly enriched (*p* = 0.0065), suggesting a potential role in environmental sensing and immune adaptation. Collectively, these findings highlight precipitation-related selective signatures at both regulatory and expression levels, offering novel insights into the mechanisms of local environmental adaptation in Chinese indigenous pigs.

### 3.5. MS4A7 as a Candidate Gene for Precipitation-Driven Local Adaptation

Genome-wide LFMM analysis identified an SNP (2_11304356_T_A) within the 11.29–11.30 Mb region of chromosome 2 that showed the strongest association with BIO16. This SNP is located in an intronic region of the *MS4A7* gene. LD analysis revealed that the top SNP exhibited low LD with neighboring loci, indicating an independent association signal. Multiple lines of selection evidence—including nucleotide diversity (π), Tajima’s D, F*_ST_*, and XP-EHH—converged at this locus, suggesting marked population differentiation and potential local adaptation driven by precipitation gradients (Figure 5a).

Genotype–environment association analysis demonstrated that individuals with the heterozygous A/T genotype had significantly higher BIO16 values compared to homozygous A/A or T/T genotypes (Figure 5b), supporting a potential regulatory role for this SNP in precipitation responsiveness. Furthermore, breed-level genotype frequency analysis of the top SNP (2_11304356_T_A) revealed that heterozygotes (A/T) were more prevalent in breeds from regions with higher precipitation, while the T/T genotype dominated in drier areas (Appendix A). This breed-level pattern reinforces the environmental relevance of the SNP and supports a role for heterozygosity in local adaptation to varying precipitation conditions (Appendix A). Chromatin state annotation across 14 porcine tissues revealed that the top SNP lies within a repressed chromatin region in the cerebral cortex, suggesting transcriptional inactivation at this site in brain tissue. Conversely, active chromatin marks were observed at the 5′UTR and exon 5 of *MS4A7* in the cortex and cerebellum, but these regions were strongly repressed in the liver (Figure 5c), indicating distinct tissue-specific regulatory patterns. Transcriptomic analysis using PigGTEx data across 35 tissues showed that *MS4A7* is moderately to highly expressed in the lung, hypothalamus, adipose tissue, spleen, and small intestine (Figure 5d), tissues central to immune, neuroendocrine, and metabolic regulation. Furthermore, data from the PigBioBank resource provide additional support for the functional role of *MS4A7*. Phenome-wide association study (PheWAS) results showed significant associations between *MS4A7* variants and reproduction-related traits, such as teat number, as well as carcass and production traits (Appendix A; Appendix A). Transcriptome-wide association study (TWAS) further revealed that *MS4A7* expression levels in multiple tissues (e.g., liver and cross-tissue panels) were significantly correlated with teat number, meat-to-fat ratio, and fatty acid composition, with the strongest signal reaching a *p*-value of 1.58 × 10^−7^ (Appendix A). This integrative evidence from chromatin state, tissue-specific expression, and large-scale genotype–phenotype datasets supports the hypothesis that *MS4A7* contributes to precipitation-driven local adaptation and simultaneously influences economically important traits in Chinese indigenous pigs.

## 4. Discussion

This study presents an integrative analysis of the genomic landscape and ecological adaptation in Chinese indigenous pig populations. For genotype–environment association, we determined a representative coordinate for each breed based on the geographic center of its historical distribution and extracted the long-term environmental mean at that location. This approach better reflects historical selective pressures compared to using contemporary sampling coordinates, and we further subdivided broadly distributed breeds (e.g., Tibetan pigs) into geographically distinct subpopulations to improve environmental matching. Principal component and ADMIXTURE analyses consistently identified the ECN group as a genetically distinct cluster, likely shaped by prolonged geographic isolation and independent domestication history [3,6]. In contrast, SCN and SWCN populations exhibited higher nucleotide diversity and Tajima’s D values. These patterns may suggest that southern pig populations experienced more complex demographic histories, such as gene flow or balancing selection, potentially shaped by their ecologically diverse habitats [39]. However, we note that Tajima’s D is sensitive to population structure, which can inflate D values even in the absence of balancing selection. Specifically, genetic divergence among subpopulations, as revealed by our PCA and ADMIXTURE analyses, can lead to elevated nucleotide diversity (π) without a corresponding increase in segregating sites, thereby biasing D estimates upward. As such, the observed positive Tajima’s D values should be interpreted with caution and considered in conjunction with other statistics and functional evidence to robustly infer selection.

The ecological basis of this genetic divergence was further supported by environmental PCA, which showed marked differences among groups along major environmental gradients, particularly temperature and precipitation seasonality. Notably, the SCN population displayed a wide distribution in the environmental PCA space, indicating broader habitat variability and higher adaptive plasticity. By integrating principal component loadings and gradient forest analysis, we identified six key environmental variables—BIO2, BIO3, BIO4, BIO8, BIO15, and BIO16—as representative predictors for downstream analyses. These variables encapsulate the core ecological gradients relevant to the habitat differentiation of local pig breeds and align with commonly applied frameworks in adaptation studies of livestock and wild species [30,40,41].

Building on this foundation, we applied a multilayered analytical framework that integrates genome-wide diversity, environmental ordination, RDA, LFMM, and Mantel tests to dissect the adaptive genetic basis of local pig populations. RDA results underscored the central role BIO4 and precipitation-related factors (BIO15 and BIO16) in shaping genomic variation, particularly highlighting stronger precipitation-related adaptation in the SWCN group. These findings align with broader patterns observed in livestock and wildlife, where environmental gradients often drive localized genomic differentiation [17,42,43]. Partial Mantel tests further demonstrated that environmental distances retained a significant association with genetic differentiation even after controlling for geographic distance, reinforcing the independent role of environmental selection in shaping adaptive divergence. Similar conclusions have been drawn in goats [44] and sheep [39], validating the utility of joint environmental and genomic analyses in disentangling signals of selection from neutral evolution [45,46].

The significantly associated SNPs with BIO16 were preferentially enriched in upstream regulatory and intergenic regions, suggesting potential roles in gene expression modulation rather than direct coding changes, which aligns with previous findings that regulatory variants contribute substantially to environmental adaptation [47,48,49]. The tissue-specific enrichment analysis further revealed that these SNPs were overrepresented in genes expressed in the lung, intestine, hypothalamus, adipose tissue, and spleen—organs critically involved in immune regulation, metabolic balance, and water homeostasis. These physiological systems are known to mediate adaptive responses to hydric stress and environmental variation [50]. For instance, lung and intestinal epithelia maintain fluid balance, while adipose and hypothalamic tissues modulate energy and stress responses [51,52]. Moreover, GO and KEGG enrichment analyses of candidate genes indicated associations with cilium organization, epithelial function, and immune-related pathways, implying that traits such as mucosal immunity, barrier integrity, and pathogen resistance may play important roles in adaptation to high-precipitation environments. Together, these findings imply that the identified SNPs may underlie regulatory mechanisms enabling local pigs to cope with spatial and seasonal changes in precipitation.

Among these candidate loci, the top SNP identified was 2_11304356_T_A, located within an intronic region of the *MS4A7* gene on chromosome 2. This SNP exhibited the strongest association with BIO16 and showed convergent evidence of selection from multiple statistics, including elevated F*_ST_*, Tajima’s D, and XP-EHH (Figure 5a). Notably, genotype–environment association analysis revealed that individuals with the A/T heterozygous genotype had significantly higher BIO16 values than both homozygotes (A/A and T/T), suggesting a potential case of overdominance or heterozygote advantage (Figure 5b). This pattern is indicative of balancing selection, which may favor heterozygous individuals in ecologically stressful environments, such as those with high precipitation and pathogen exposure. Breed-level genotype frequency analysis further revealed that A/T heterozygotes were more prevalent in breeds from regions with higher precipitation, whereas the T/T genotype dominated in drier areas (Figure 5c), reinforcing the environmental relevance of this SNP. Regulatory annotation showed that while 2_11304356_T_A lies within a repressive chromatin region in the cortex, active regulatory elements were identified at the 5′UTR and exon 5 in multiple peripheral tissues, including adipose tissue and cerebellum (Figure 5c). Furthermore, transcriptomic data from PigGTEx demonstrated moderate to high expression of *MS4A7* in tissues closely related to immune and barrier functions, such as the lung, small intestine, spleen, hypothalamus, and adipose tissue (Figure 5d) suggesting an immunoregulatory role under humid, pathogen-rich conditions [53,54]. Heterozygote advantage at this locus may result from the combined functional benefits of both alleles, potentially broadening pathogen recognition repertoires or balancing immune activation with metabolic costs [55]. Similar cases of overdominance have been documented in the MHC loci of vertebrates, where increased allelic diversity enhances resistance to a wider range of pathogens [56]. The co-occurrence of balancing selection signals (e.g., heterozygote advantage, positive Tajima’s D) and directional selection signals (e.g., high XP-EHH) at this locus suggests a complex adaptive scenario. It is plausible that balancing selection maintains genetic diversity at the population level, while recent local directional selection acts on specific alleles in populations exposed to high-BIO16 environments. Collectively, these results underscore *MS4A7* as a compelling candidate gene underlying precipitation-mediated environmental adaptation in Chinese indigenous pigs. While the causality between *MS4A7* function and precipitation adaptation remains to be experimentally validated, future work incorporating gene expression assays and genome editing under contrasting precipitation environments is warranted.

This study integrates whole-genome variation, genotype–environment association, and multilayer functional annotation to systematically uncover the genomic basis of local adaptation. Our findings highlight BIO16 as a key environmental driver and provide novel insights into the evolutionary dynamics of livestock adaptation, with implications for climate-resilient conservation and breeding strategies. In practical terms, the adaptive loci identified here can serve as molecular markers to prioritize indigenous breeds for conservation, thereby preserving adaptive diversity under climate change [8]. Moreover, incorporating such loci into genomic selection programs may improve resilience-related traits such as pathogen resistance and metabolic stability, supporting sustainable pig breeding [57]. Finally, these findings provide a valuable genomic resource that can be introgressed into commercial populations to enhance global pig improvement. However, we acknowledge that our sampling lacked sufficient representation from arid northwestern regions of China, which may limit the generalizability of our findings across all climatic zones. Future efforts should aim to incorporate additional samples from these underrepresented areas to enhance the ecological coverage and analytical resolution of local adaptation studies.

## 5. Conclusions

This study reveals the genomic signatures of local climate adaptation in Chinese indigenous pigs, with BIO16 identified as a key environmental factor. By integrating genome-wide association analysis, environmental modeling, and functional annotation, we identified candidate loci and regulatory mechanisms associated with climate adaptation, including the gene *MS4A7*. These findings improve our understanding of environmental adaptation in livestock and provide a valuable foundation for conservation and climate-resilient breeding programs.

## Figures and Tables

**Figure 1 animals-15-02412-f001:**
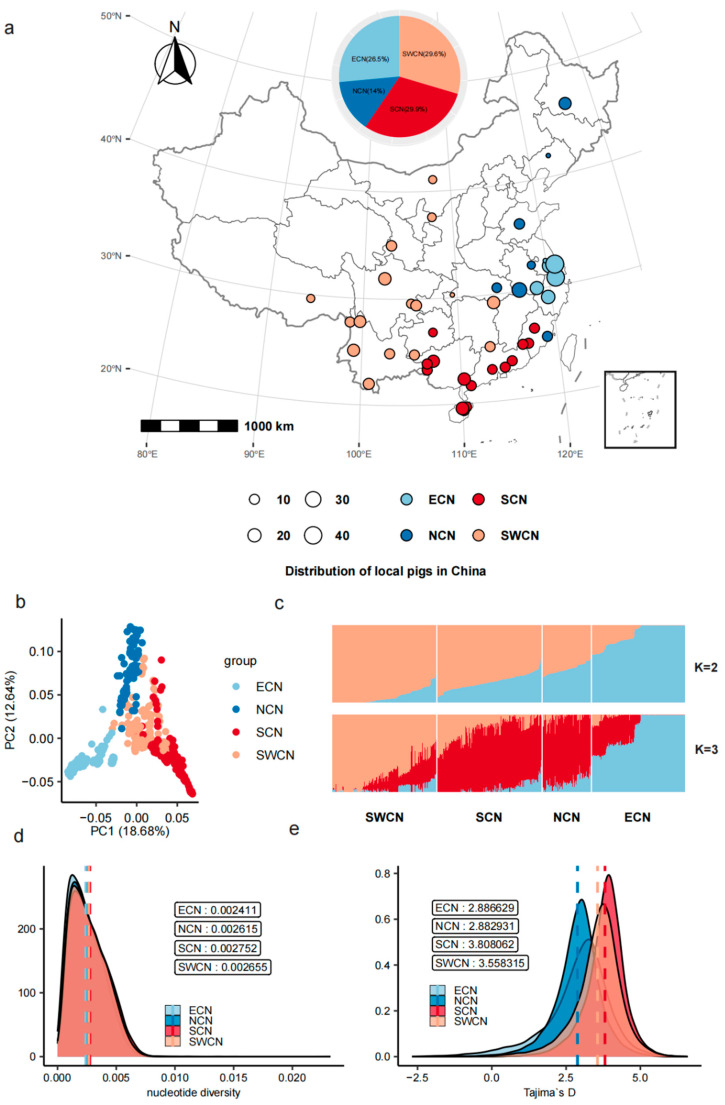
Population structure and genetic diversity of Chinese indigenous pigs. (**a**) Geographic distribution of 46 indigenous pig populations grouped into four regions: ECN (blue), NCN (light blue), SCN (red), and SWCN (orange). Circle size reflects sample size. (**b**) PCA of 578 individuals based on genome-wide SNPs; PC1 and PC2 explain 18.68% and 12.64% of variation, respectively. (**c**) ADMIXTURE plots at K = 2 and K = 3, illustrating population structure and ancestry components. (**d**) Nucleotide diversity (π) distributions across population groups. (**e**) Distribution of Tajima’s D values by group, with dashed lines indicating group means.

**Figure 2 animals-15-02412-f002:**
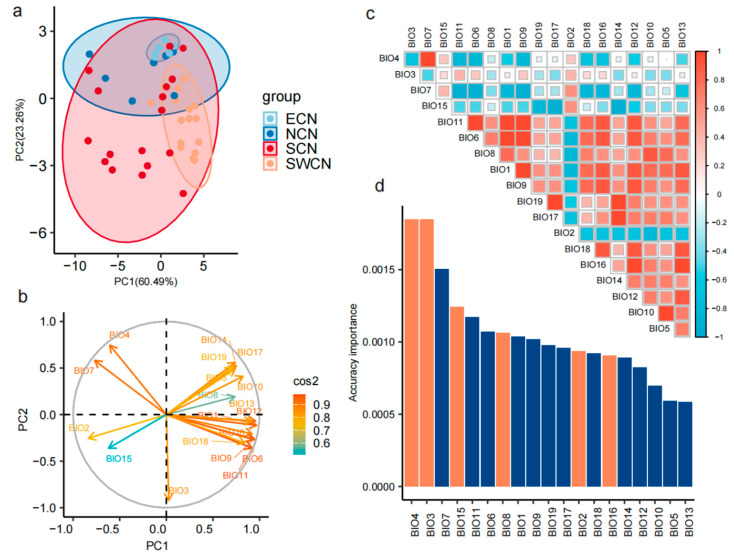
Environmental PCA and key variable selection. (**a**) Biplot of the principal component analysis (PCA) based on 19 bioclimatic variables. Arrows indicate the loading directions of environmental variables in the PC1–PC2 space, and arrow color denotes the cos^2^ value (i.e., the contribution of each variable to the principal components), ranging from blue-green (low contribution) to red (high contribution). (**b**) Distribution of 46 Chinese indigenous pig populations in the environmental PCA space defined by PC1 and PC2. Each point represents a population, colored by genetic group: ECN (blue), NCN (light blue), SCN (red), and SWCN (orange). Ellipses indicate 95% confidence intervals of population clustering. (**c**) Pairwise Pearson correlation matrix among the 19 bioclimatic variables. Color intensity and square size indicate the strength and direction of correlation (red = positive, blue = negative). (**d**) Bar plot showing the ranked importance of each environmental variable in the PCA. Orange bars denote the variables retained for subsequent analyses.

**Figure 3 animals-15-02412-f003:**
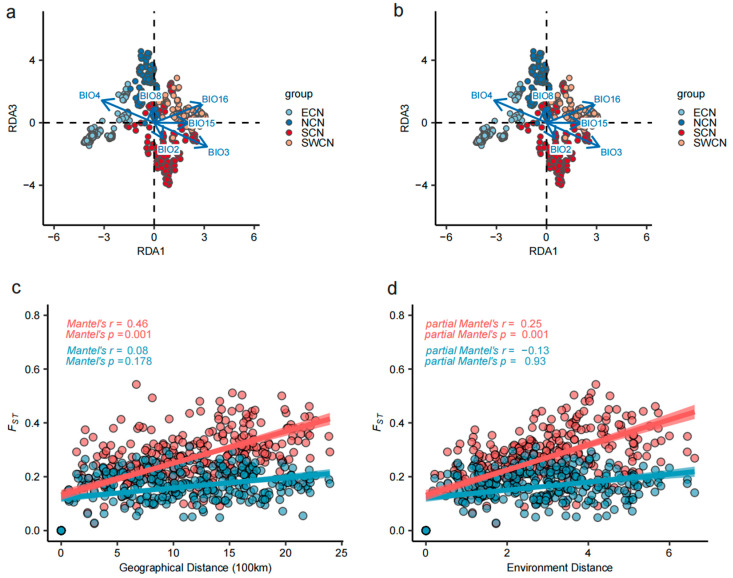
Redundancy analysis (RDA) and distance-based correlation reveal environmental influence on genomic variation in Chinese indigenous pig populations. (**a**,**b**) RDA results showing the relationship between genomic variation and environmental variables across 578 individuals from 46 local pig populations. Each point represents an individual, colored by population group (ECN, NCN, SCN, SWCN). Blue arrows indicate the direction and strength of the six selected environmental variables in the constrained ordination space. The projections of arrows on RDA axes reflect the relative contribution of each environmental factor. (**c**,**d**) Correlation between pairwise FST and geographic or environmental distance. (**c**) Relationship between F*_ST_* and geographic distance among populations based on background SNPs (blue) and LFMM-identified environment-associated SNPs (pink). The top-left inset shows Mantel correlation coefficients and significance values. (**d**) Relationship between FST and environmental distance after controlling for geographic distance. Partial Mantel test results are shown in the top-left corner. Significant correlations for LFMM SNPs but not background SNPs suggest an independent role of environmental selection in shaping population differentiation.

**Figure 4 animals-15-02412-f004:**
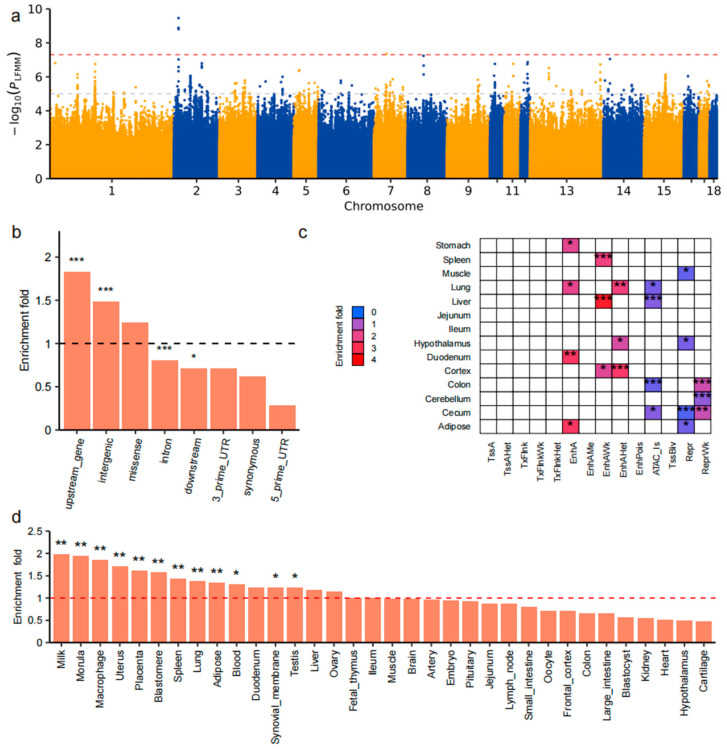
Multilayer annotation and enrichment analysis of SNPs significantly associated with BIO16. (**a**) Manhattan plot of genome-wide LFMM association analysis for BIO16. The *y*-axis represents −log_10_(*p*-value); the red dashed line denotes the genome-wide significance threshold (5 × 10^−8^), and the gray dashed line marks the suggestive threshold (1 × 10^−5^). (**b**) Enrichment of significant SNPs across genomic functional categories. *** *p* < 0.001, * *p* < 0.05; dashed line indicates enrichment fold = 1. (**c**) Heatmap of chromatin state enrichment across tissues. Each cell represents a combination of chromatin state (rows) and tissue (columns). Red indicates positive enrichment; blue indicates negative enrichment. *** *p* < 0.001, ** *p* < 0.01, * *p* < 0.05. (**d**) Enrichment analysis of significant SNPs in the top 1000 tissue-specific genes from 34 tissues. The *y*-axis indicates enrichment fold; ** *p* < 0.01, * *p* < 0.05; dashed line indicates enrichment fold = 1.

**Figure 5 animals-15-02412-f005:**
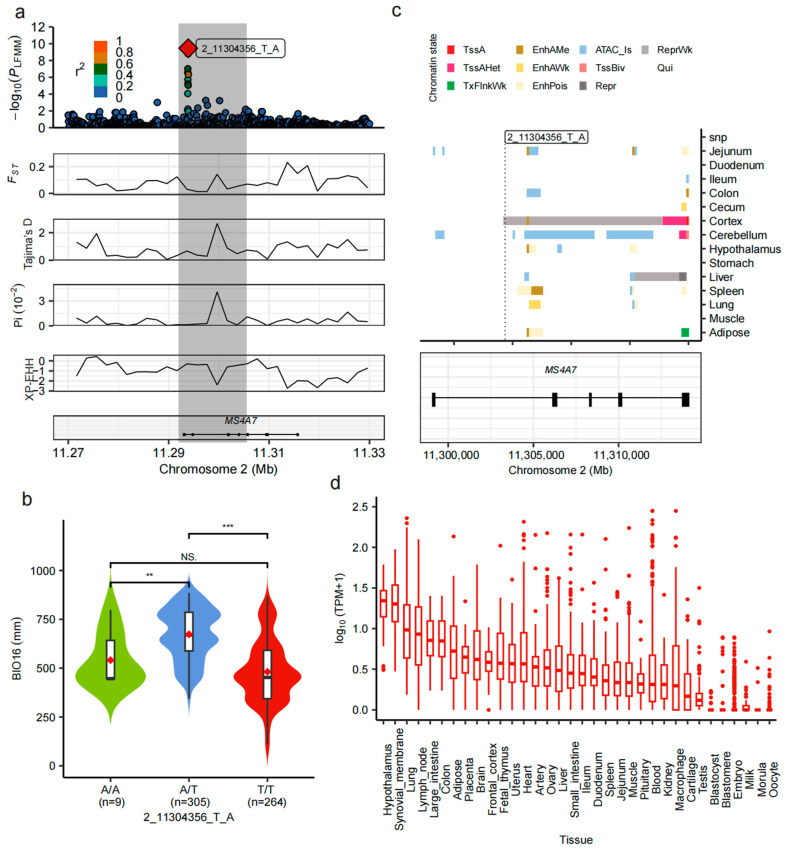
Multilayered annotation of the *MS4A7* gene region associated with BIO16. (**a**) Local annotation of the *MS4A7* region. The top panel shows the local association and LD plot (r^2^ indicating linkage strength); the bottom panels display the distributions of π, Tajima’s D, F*_ST_*, and XP-EHH, highlighting a strong selective signal centered at 2_11304356_T_A. (**b**) Distribution of BIO16 values among individuals with different genotypes. A/T carriers show significantly higher BIO16 values than A/A and T/T individuals, *** *p* < 0.001, ** *p* < 0.01, ns not significant. (**c**) Chromatin state and tissue-specific regulatory annotation. The upper panel presents regulatory activity (e.g., enhancers, open chromatin) across different tissues in the *MS4A7* region; the lower panel shows gene structure and the position of the top SNP relative to exons and UTRs. (**d**) Expression profile of *MS4A7* across 35 pig tissues. TPM data indicate moderate to high expression levels in the lung, hypothalamus, small intestine, spleen, and adipose tissue, suggesting diverse functional roles.

## Data Availability

All the data necessary to evaluate the conclusions of this paper are included within the main manuscript and/or the Appendix A. The 578 resequenced datasets analyzed in this study were derived from our previously published pig genomics reference panel (PGRP v1) [18]). Detailed information on these datasets is provided in Appendix A.

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
