# Peer review of "Local Climate Adaptation in Chinese Indigenous Pig Genomes"

_animals, 2025, doi:10.3390/ani15162412_

Round 1
Reviewer 1 Report
Comments and Suggestions for Authors
This manuscript systematically investigates the environmental adaptation and genetic basis of Chinese indigenous pigs by integrating multilayer approaches, including whole-genome resequencing data, environmental variable analysis, genotype-environment association (GEA), and functional annotation. The study provides valuable theoretical for the conservation of local pig germplasm resources and climate-resilient breeding strategies.
Major Comments:
- The authors employed the LFMM model for GEA analysis, which is an appropriate choice. However, the description of the methods section is overly simplified and lacks critical details regarding parameter settings. The performance of LFMM heavily depends on the correct selection of the latent factor K, which is used to adjust for population structure. The authors did not specify how they determined the optimal K in section 2.4. Please add the specific method used for selecting K and the final K value applied, as this is crucial for ensuring the reliability of the results.
- In the genotypic data processing section (section 2.1, line 80), the authors used a filtering criterion of MAF < 0.1. This threshold is too stringent for population-level studies. Variants associated with local adaptation might have high frequencies in one or a few populations but low frequencies across the entire dataset. Setting MAF > 0.1 may exclude many rare variants that are critical for local adaptation. Please justify the use of such a high MAF threshold or consider using a more lenient cutoff (e.g., MAF > 0.01 or 0.05) and re-analyze to evaluate the impact on results.
- The results section (section 3.1, lines 181-185) reports that Tajima's D values are positive across all four population groups, interpreted as evidence of balancing selection or bottlenecks. While this is a standard interpretation, in the presence of significant population structure (as shown by PCA and ADMIXTURE), the calculation of Tajima's D across mixed populations can bias the results toward positive values. The authors should mention that population structure might be an important factor contributing to the positive Tajima's D, which would make the interpretation more rigorous.
- Figure 5b presents very interesting results, showing that heterozygous individuals (A/T) at BIO16 (precipitation in the wettest season) have significantly higher environmental values than homozygous individuals (A/A and T/T). This strongly suggests overdominance or balancing selection. However, in the discussion, the authors only briefly mention its adaptive significance. How does this heterozygote advantage relate to adaptation to high precipitation environments? For instance, could it be linked to immune system trade-offs, such as broader pathogen resistance in wetter, pathogen-rich environments? Does the signal of balancing selection (e.g., heterozygote advantage) align with other signals like XP-EHH? Since XP-EHH typically detects directional selection, it would be valuable to discuss whether these signals are compatible or represent different adaptive processes. A more in-depth discussion on this point is recommended.
Minor Comments:
- The abstract (line 25) mentions integrating "future climate projections," but no such analyses are presented in the main text. Please either include the future climate scenario analyses or revise the abstract accordingly.
- In Figure 2b, the legend (line 225) states "Distribution of 45 Chinese indigenous pig populations," but the methods and abstract mention 46 breeds. Please verify and correct this inconsistency.
- In section 3.1 (line 171), PCA used 155,686 SNPs, whereas in section 3.3 (line 233), RDA used 226,394 LD-pruned SNPs. Why are different SNP sets used for population structure and for environmental association analyses? To ensure comparability, it is recommended to use the same high-quality, LD-pruned SNP set across all analyses or clarify the rationale for different selections.
- Line 278: The notation "=1.4p, p=3.67 × 10-13" contains a typo; the first "p" should be deleted.
- Multiple instances of "Tajima`s D" should be corrected to "Tajima's D" (standard apostrophe).
- The window size for candidate gene identification (±25 kb) should be justified or explained.
- In line 417, the manuscript states "568 resequenced datasets," but the abstract mentions 578 individuals. Please verify and clarify the dataset details.
- The DOI formats for references [2] and [11] are incorrect ("doi:doi:"). Please correct all DOI formats to standard.
Author Response
Comments: This manuscript systematically investigates the environmental adaptation and genetic basis of Chinese indigenous pigs by integrating multilayer approaches, including whole-genome resequencing data, environmental variable analysis, genotype-environment association (GEA), and functional annotation. The study provides valuable theoretical for the conservation of local pig germplasm resources and climate-resilient breeding strategies.
Response: We sincerely thank the reviewer for their positive evaluation of our manuscript. We are grateful for the recognition of our integrative approach combining whole-genome resequencing, environmental variable analysis, genotype–environment association, and functional annotation to explore the genetic basis of environmental adaptation in Chinese indigenous pigs. We are also pleased that the reviewer found our study to provide valuable insights for the conservation of local pig germplasm resources and climate-resilient breeding strategies. We have carefully addressed all the comments and suggestions, and we believe the revisions have strengthened the clarity and scientific rigor of the manuscript.
Comments 1: The authors employed the LFMM model for GEA analysis, which is an appropriate choice. However, the description of the methods section is overly simplified and lacks critical details regarding parameter settings. The performance of LFMM heavily depends on the correct selection of the latent factor K, which is used to adjust for population structure. The authors did not specify how they determined the optimal K in section 2.4. Please add the specific method used for selecting K and the final K value applied, as this is crucial for ensuring the reliability of the results.
Response 1:We thank the reviewer for pointing out this important issue. In the revised manuscript, we have added details regarding the selection of the latent factor K in LFMM analysis (Methods, Section 2.4,lines:129-134). Briefly, we first performed PCA, which revealed clear clustering into four geographic groups (Fig. 1b). We then conducted ADMIXTURE analysis from K = 2 to K = 10. Although the cross-validation error curve did not show a sharp elbow point, the error began to decrease slowly from K = 4. Considering population structure, domestication history, and geographic distribution of Chinese indigenous pigs, we selected K = 4 as the latent factor. This parameter setting is now explicitly stated in the revised manuscript.
Comments 2: In the genotypic data processing section (section 2.1, line 80), the authors used a filtering criterion of MAF < 0.1. This threshold is too stringent for population-level studies. Variants associated with local adaptation might have high frequencies in one or a few populations but low frequencies across the entire dataset. Setting MAF > 0.1 may exclude many rare variants that are critical for local adaptation. Please justify the use of such a high MAF threshold or consider using a more lenient cutoff (e.g., MAF > 0.01 or 0.05) and re-analyze to evaluate the impact on results.
Response 2: We thank the reviewer for this insightful comment. Our initial decision to use MAF > 0.1 was based on previous studies (e.g., Sang et al., Nat. Commun. 2022, https://doi.org/10.1038/s41467-022-34206-8), which applied this threshold to reduce false positives and filter out low-confidence variants in GEA analysis using LFMM. To directly address the reviewer’s concern, we re-performed the GEA using a more lenient threshold of MAF > 0.05, focusing on BIO16 as the target environmental variable. The resulting Manhattan plots (Supplementary Fig. S1a–b) revealed highly consistent peak signals between the two MAF thresholds. Moreover, –log₁₀(P) values showed a strong Pearson correlation (r = 0.992; Fig. S1c). These results suggest that our GEA findings are robust to different MAF thresholds. This comparison and its rationale have been added to Section 3.4,lines 313-320 and the Supplementary Materials.
Comments 3: The results section (section 3.1, lines 181-185) reports that Tajima's D values are positive across all four population groups, interpreted as evidence of balancing selection or bottlenecks. While this is a standard interpretation, in the presence of significant population structure (as shown by PCA and ADMIXTURE), the calculation of Tajima's D across mixed populations can bias the results toward positive values. The authors should mention that population structure might be an important factor contributing to the positive Tajima's D, which would make the interpretation more rigorous.
Response 3: We appreciate the reviewer’s thoughtful observation. In the revised Results (Section 4,lines:434-445), we now explicitly acknowledge that Tajima's D can be upwardly biased in the presence of population structure, even without true balancing selection. Specifically, divergence among subpopulations—as shown by PCA and ADMIXTURE—can elevate nucleotide diversity (π) without a proportional increase in segregating sites, thereby inflating D values. Accordingly, we now state that the observed positive Tajima’s D values should be interpreted with caution and in conjunction with functional evidence and additional statistics.
Comments 4: Figure 5b presents very interesting results, showing that heterozygous individuals (A/T) at BIO16 (precipitation in the wettest season) have significantly higher environmental values than homozygous individuals (A/A and T/T). This strongly suggests overdominance or balancing selection. However, in the discussion, the authors only briefly mention its adaptive significance. How does this heterozygote advantage relate to adaptation to high precipitation environments? For instance, could it be linked to immune system trade-offs, such as broader pathogen resistance in wetter, pathogen-rich environments? Does the signal of balancing selection (e.g., heterozygote advantage) align with other signals like XP-EHH? Since XP-EHH typically detects directional selection, it would be valuable to discuss whether these signals are compatible or represent different adaptive processes. A more in-depth discussion on this point is recommended.
Response 4: We thank the reviewer for highlighting this interesting result and for suggesting a deeper biological interpretation. In the revised Discussion (lines 488–522), we now propose that the observed heterozygote advantage at SNP 2_11304356_T_A may confer fitness benefits in high-precipitation environments. This could result from immune trade-offs, where heterozygotes at the MS4A7 locus maintain broader pathogen resistance or more balanced immune activation. This is consistent with the expression profile of MS4A7 in immune and barrier tissues such as the lung, intestine, and spleen.
We also discuss the potential compatibility between balancing selection (heterozygosity, Tajima’s D) and XP-EHH (directional selection), suggesting a scenario of temporally or spatially heterogeneous selection. For instance, a long-term balancing regime may have transitioned into recent directional selection in specific ecological contexts. We thank the reviewer for encouraging us to elaborate on this point.
Minor Comments:
Comments 5: The abstract (line 25) mentions integrating "future climate projections," but no such analyses are presented in the main text. Please either include the future climate scenario analyses or revise the abstract accordingly.
Response 5: We agree with the reviewer. The phrase “future climate projections” has been removed from the abstract to reflect the current scope of analysis.
Comments 6: In Figure 2b, the legend (line 225) states "Distribution of 45 Chinese indigenous pig populations," but the methods and abstract mention 46 breeds. Please verify and correct this inconsistency.
Response 6: Thank you for catching this inconsistency. The correct number is 46 pig populations. We have corrected the legend in Figure 2b accordingly.
Comments 7: In section 3.1 (line 171), PCA used 155,686 SNPs, whereas in section 3.3 (line 233), RDA used 226,394 LD-pruned SNPs. Why are different SNP sets used for population structure and for environmental association analyses? To ensure comparability, it is recommended to use the same high-quality, LD-pruned SNP set across all analyses or clarify the rationale for different selections.
Response 7: We appreciate the reviewer for pointing this out. This was a typographical error in the original manuscript. We confirm that both PCA and RDA were conducted using the same set of 226,394 LD-pruned SNPs. The SNP count has been corrected in the revised text (Section 3.1, line;211).
Comments 8: Line 278: The notation "=1.4p, p=3.67 × 10⁻¹³" contains a typo; the first "p" should be deleted.
Response 8: Corrected. The sentence now reads “fold enrichment = 1.4, p = 3.67 × 10⁻¹³.”
Comments 9: Multiple instances of "Tajima`s D" should be corrected to "Tajima's D" (standard apostrophe).
Response 9: All instances have been corrected accordingly.
Comments 10: The window size for candidate gene identification (±25 kb) should be justified or explained.
Response 10: Thank you for the suggestion. We assessed genome-wide linkage disequilibrium (LD) decay across the 46 pig populations and found that the average r² plateaued around 40–50 kb (Supplementary Fig. S2). Based on this result, a ±25 kb window was chosen to capture relevant regulatory variants while limiting noise. This justification is now included in Section 3.4 ,lines:321-328.
Comments 11: In line 417, the manuscript states "568 resequenced datasets," but the abstract mentions 578 individuals. Please verify and clarify the dataset details.
Response 11: Thank you. The correct number is 578 individuals. The inconsistency has been corrected.
Comments 12:The DOI formats for references [2] and [11] are incorrect ("doi:doi:"). Please correct all DOI formats to standard.
Response 12:We have thoroughly checked and corrected all DOI formatting errors in the manuscript.

Reviewer 2 Report
Comments and Suggestions for Authors
This manuscript addresses the genetic basis of local adaptation in Chinese indigenous pig breeds, an important topic in animal genomics. The authors applied an integrative framework combining whole genome resequencing, genotype environment association analyses, and climate modeling to explore adaptation to precipitation gradients, particularly BIO16. The study presents a certain level of novelty and forward-looking perspective, and the topic is well aligned with the scope of Animals. While the manuscript is overall sound and suitable for publication, several comments in data interpretation, methodological details, and logical coherence require minor revisions to enhance clarity and scientific rigor.
1.Given the prominent role of BIO16 across multiple analyses, it is recommended to include relevant ecological or climatic background information in the Introduction, particularly regarding its potential influence on local adaptation, to better justify the biological significance of precipitation in the wettest quarter and enhance the logical coherence and scientific rigor of the manuscript.
2.Line 113: Please provide justification for the P-value threshold used (P < 1 × 10-5) and include a supporting reference.
3.Lines 112–113: These results should not be presented within the Materials and Methods section. Please move them to the appropriate location in the Results section.
4.Lines 122–125: The description is redundant as it is already covered in the Results section and can be omitted.
5.Lines 150–151: Similarly, associated findings (e.g., number of genes identified) should appear in the results section. And reference 34 (for BEDTools) should be cited directly within the method description as "BEDTools v2.25.0 [34]" rather than at the end of a result sentence.
6.Lines 236–241: The redundancy analysis (RDA) results lack quantitative descriptions of significance and the proportion of variance explained by each axis. Please provide these statistics to improve reproducibility and credibility.
7.Lines 246–249: The reported correlation coefficients (r) and P-values are inconsistent with those in Figures 3c–d. Please verify and revise accordingly.
Additionally, Figures 3c–d are not cited at the appropriate point in the text, and Font formatting in lines 252–267 is inconsistent, please revise.
8.Lines 278–279: The textual description does not match Figure 4b. The figure shows that SNPs were significantly enriched in intergenic region(~1.4-fold) rather than intron. Please correct this discrepancy, also revise line380.
9.Figure 4c: Please revise the color legend. Negative fold enrichment values in blue should also be clearly labeled in the figure for interpretability.
Author Response
Comments: This manuscript addresses the genetic basis of local adaptation in Chinese indigenous pig breeds, an important topic in animal genomics. The authors applied an integrative framework combining whole genome resequencing, genotype environment association analyses, and climate modeling to explore adaptation to precipitation gradients, particularly BIO16. The study presents a certain level of novelty and forward-looking perspective, and the topic is well aligned with the scope of Animals. While the manuscript is overall sound and suitable for publication, several comments in data interpretation, methodological details, and logical coherence require minor revisions to enhance clarity and scientific rigor.
Response: We thank the reviewer for the positive feedback and valuable suggestions. We are pleased that our study on the genetic basis of local adaptation in Chinese indigenous pig breeds is considered innovative and forward-looking, aligning well with the scope of Animals. We have carefully addressed each of the reviewer’s comments regarding data interpretation, methodological details, and logical coherence. All suggested changes have been implemented in the revised manuscript, and we believe these improvements further enhance its scientific rigor and clarity. Below are our detailed responses.
Comments1: Given the prominent role of BIO16 across multiple analyses, it is recommended to include relevant ecological or climatic background information in the Introduction, particularly regarding its potential influence on local adaptation, to better justify the biological significance of precipitation in the wettest quarter and enhance the logical coherence and scientific rigor of the manuscript.
Response1: We thank the reviewer for this insightful suggestion. In the revised manuscript, we have added relevant ecological and climatic background information in the Introduction (lines 73–85) to better justify the biological significance of BIO16, particularly its potential influence on local adaptation. This addition enhances the logical coherence and scientific rigor of the manuscript.
Comments 2: Line 113: Please provide justification for the P-value threshold used (P < 1 × 10-5) and include a supporting reference.
Response2: We thank the reviewer for this important comment. In our LFMM-based genotype–environment association (GEA) analysis, we adopted a significance threshold of P < 1 × 10⁻⁵ to identify SNPs suggestively associated with environmental gradients. This threshold corresponds approximately to |z-score| > 4, which has been used in previous methodological studies to control for false positives under a stringent genome-wide scan. Specifically, Frichot et al. (2013) stated that “the cutoff |z| > 4 corresponds to P < 10⁻⁵ obtained after applying a Bonferroni correction for a type I error α = 0.01 and L of order 10³ loci” (Mol. Biol. Evol., 30(7):1687–1699. https://doi.org/10.1093/molbev/mst063). This conservative threshold balances discovery power and false-positive control in large-scale datasets. This rationale and citation have been included in the revised Methods section (Section 2.4, lines 139–142).
Comments 3: Lines 112–113: These results should not be presented within the Materials and Methods section. Please move them to the appropriate location in the Results section.
Response 3: We have moved this content to the Results section (Section 3.1, lines 207–211) as suggested.
Comment 4: Lines 122–125: The description is redundant as it is already covered in the Results section and can be omitted.
Response 4: We have removed the redundant description from the Methods section and retained the relevant content in the Results section (Section 3.2, lines 251–253).
Comment 5: Lines 150–151: Similarly, associated findings (e.g., number of genes identified) should appear in the results section. And reference 34 (for BEDTools) should be cited directly within the method description as "BEDTools v2.25.0 [34]" rather than at the end of a result sentence.
Response 5: We have moved the relevant results to the Results section and revised the Methods section to cite “BEDTools v2.25.0 [34]” directly within the description(Section 2.7, line 178).
Comments 6: Lines 236–241: The redundancy analysis (RDA) results lack quantitative descriptions of significance and the proportion of variance explained by each axis. Please provide these statistics to improve reproducibility and credibility.
Response 6: We appreciate the reviewer’s suggestion. We performed permutation tests for each environmental variable, and all six retained predictors showed statistically significant associations with genomic variation (P < 0.01). The first three RDA axes explained 44.5%, 18.8%, and 13.3% of the constrained variance, respectively, totaling 76.6% of the explained variation. These results have been added to the Results section (lines 270–278) for improved interpretability and rigor.
Comments 7 : Lines 246–249: The reported correlation coefficients (r) and P-values are inconsistent with those in Figures 3c–d. Please verify and revise accordingly.Additionally, Figures 3c–d are not cited at the appropriate point in the text, and Font formatting in lines 252–26 tent, please revise.
Response 7 : We have re-checked the correlation coefficients and P-values in Figures 3c–d and corrected the text to ensure consistency. The citation for Figures 3c–d has been moved to line 289, where the results are first described, and font formatting issues between lines 279–288 have been corrected.
Comment 8: Lines 278–279: The textual description does not match Figure 4b. The figure shows that SNPs were significantly enriched in intergenic region(~1.4-fold) rather than intron. Please correct this discrepancy, also revise line380.
Response 8: We thank the reviewer for pointing this out. The description has been corrected to state that SNPs were significantly enriched in intergenic regions (~1.4-fold), consistent with Figure 4b. Corrections have been made at line 334 and line 463 of the revised manuscript.
Comment 9 : Figure 4c: Please revise the color legend. Negative fold enrichment values in blue should also be clearly labeled in the figure for interpretability.
Response 9: We have revised the color legend in Figure 4c to clearly label the negative fold enrichment values shown in blue, improving figure interpretability. The updated figure is included in the revised manuscript.
Reviewer 3 Report
Comments and Suggestions for Authors
Dear Authors,
Manuscript ID: animals-3782206
Local Climate Adaptation in Chinese Indigenous Pig Genomes
Chinese indigenous pigs represent a globally important genetic pool with high phenotypic diversity, adaptability, and rich evolutionary history. The use of whole-genome resequencing and population genomics tools provides deep insights into selection signatures, local adaptation, and evolutionary processes, which are relevant to conservation and breeding programs worldwide. While this manuscript has the potential to be an excellent academic reference, several issues must be addressed to improve its completeness before assessing its suitability for publication, as outlined below.
Major comments:
- What specific phenotypic traits (e.g., heat tolerance, disease resistance, fat deposition) are associated with the local adaptation signals identified in the genomic regions under selection?
- Could the authors provide more functional insights or biological relevance of the key candidate genes identified, particularly how these genes contribute to local adaptation in different environments?
- How were the pig populations selected for this study? Were there any limitations in geographical coverage or breed diversity that might bias the detection of selection signals
- Have any of the candidate genes or loci identified in this study been validated experimentally or supported by expression data or GWAS in pigs?
- How can the genomic findings from this study be practically applied to conservation strategies or breeding programs for Chinese indigenous pigs or global pig improvement efforts?
Best Regards
Reviewer
Author Response
Comment: Chinese indigenous pigs represent a globally important genetic pool with high phenotypic diversity, adaptability, and rich evolutionary history. The use of whole-genome resequencing and population genomics tools provides deep insights into selection signatures, local adaptation, and evolutionary processes, which are relevant to conservation and breeding programs worldwide. While this manuscript has the potential to be an excellent academic reference, several issues must be addressed to improve its completeness before assessing its suitability for publication, as outlined below.
General Response:
We thank the reviewer for the positive evaluation of our work and the constructive comments. We are pleased that our study is considered relevant, novel, and aligned with the scope of Animals. We have carefully addressed each of the points raised, revised the manuscript accordingly, and believe these changes have improved both the clarity and the scientific rigor of our work. Our detailed responses are as follows.
Major comments:
Comment 1: What specific phenotypic traits (e.g., heat tolerance, disease resistance, fat deposition) are associated with the local adaptation signals identified in the genomic regions under selection?
Response 1: We thank the reviewer for this insightful suggestion. In the revised manuscript, we have added a brief discussion of potential phenotypic traits associated with local adaptation signals. Based on GO and KEGG enrichment analyses of candidate genes associated with BIO16, we identified significant enrichment in biological processes such as cilium organization, epithelial structure regulation, and immune-related signaling pathways (e.g., C-type lectin receptor signaling). These findings suggest that traits related to mucosal immunity, epithelial barrier function, and environmental stress response—particularly under high-precipitation conditions—may play critical roles in the local adaptation of Chinese indigenous pigs. This discussion has been added to the revised manuscript (Discussion, lines 473–479,480-514).
Comment 2:Could the authors provide more functional insights or biological relevance of the key candidate genes identified, particularly how these genes contribute to local adaptation in different environments?
Response 2: We thank the reviewer for this insightful question. To illustrate the biological relevance of our findings, we visualized the genotype distribution of the top SNP (2_11304356_T_A) across 46 indigenous pig breeds (Supplementary Figure S3). Each bar represents genotype percentages (A/A, A/T, T/T) for a breed, while the black line shows the corresponding BIO16 value. The pattern reveals a clear trend: the heterozygous genotype (A/T) is more frequent in breeds from high-precipitation regions, whereas the T/T genotype predominates in drier regions, supporting a genotype–environment association and possible heterozygote advantage in humid, pathogen-rich environments. This result is now included in the revised Results section (3.5, lines 381–386).
We have also expanded the Discussion (lines 497–515) to elaborate on the biological function of the top candidate gene, MS4A7, which is expressed in immune- and barrier-related tissues (e.g., lung, intestine, spleen) that are critical for host defense under high-humidity conditions. While we have made our best effort to explore how key genes contribute to local adaptation, we acknowledge that the lack of breed-specific transcriptomic data currently limits our ability to fully characterize gene regulation mechanisms, which will be addressed in future research.
Comment 3:How were the pig populations selected for this study? Were there any limitations in geographical coverage or breed diversity that might bias the detection of selection signals.
Response 3: We selected 46 Chinese indigenous pig breeds with well-documented geographic origins, representing major ecological zones such as southeastern, southwestern, central-southern, and northern China. These breeds encompass the primary genetic lineages and ecological distributions of Chinese native pigs. While we aimed to capture a broad geographic and environmental gradient, we acknowledge that arid northwestern regions are underrepresented. Future studies will expand sampling in these areas to improve representativeness and analytical resolution. This information has been added to the Discussion (lines 527–531).
Comment 4:Have any of the candidate genes or loci identified in this study been validated experimentally or supported by expression data or GWAS in pigs?
Response 4: We thank the reviewer for this important question. To evaluate the biological relevance of MS4A7, we referred to publicly available data in the PigBioBank database (https://pigbiobank.farmgtex.org/).
Expression data: Transcriptomic profiles indicate that MS4A7 is moderately to highly expressed in immune- and barrier-related tissues such as lung, intestine, spleen, and lymph nodes (Fig. 5d), supporting its role in host–pathogen interactions under humid conditions.
PheWAS: Genetic variation at the MS4A7 locus is significantly associated with teat number (Supplementary Figure S4, Supplementary Table 8).
TWAS: MS4A7 expression in multiple tissues (e.g., liver, cross-tissue panels) is significantly associated with teat number, meat-to-fat ratio, and margaric acid content, with the strongest association reaching p = 1.58 × 10⁻⁷ (Supplementary Table 9).
These independent datasets support the functional relevance of MS4A7 in both environmental adaptation and economically important traits. This evidence has been added to the Results section (lines 393–404) and detailed in the Supplementary Materials.
Comment 5:How can the genomic findings from this study be practically applied to conservation strategies or breeding programs for Chinese indigenous pigs or global pig improvement efforts?
Response 5: We thank the reviewer for this valuable suggestion. In the revised Discussion (lines 518–527), we emphasize that adaptive loci identified in this study can:Serve as molecular markers to guide conservation priorities and preserve adaptive diversity under climate change; Be incorporated into genomic selection programs to improve resilience-related traits such as pathogen resistance and metabolic stability; Be introgressed into commercial populations to enhance global pig improvement.
Reviewer 4 Report
Comments and Suggestions for Authors
The manuscript presents a study on the genomic adaptation of Chinese indigenous pig breeds to climatic conditions. The authors employ an approach that integrates whole-genome data, environmental variables, various genotype-environment association (GEA) methods, and multi-layered functional annotation. The study identifies precipitation (BIO16) as a key factor in selection and provides compelling evidence for the role of the MS4A7 gene in adaptation. This work is significant for understanding the mechanisms of local adaptation in livestock and can serve as a basis for developing programs for the conservation of genetic resources and the creation of climate-resilient breeds.
Overall, the manuscript is clearly written, the methodology is sound, and the conclusions are well-supported by the results. However, there are several points that require clarification or further discussion to make the article even more convincing.
1) In Section 2.2, it is stated that "one representative latitude-longitude coordinate was selected for each breed" to reflect its long-term local adaptation. This is a critically important point, as the accuracy of GEA analysis directly depends on the accuracy of genotype-environment linkage. The ranges of indigenous breeds can be quite extensive. What was the criterion for selecting this single point? Was it the geographic center of the historical range of the breed, the location of a specific farm from which the samples were taken, or some other parameter? The methodology for selecting coordinates should be described in detail. It would also be useful to discuss in the "Discussion" section the potential limitation associated with this simplification and assess how possible inaccuracies might affect the results.
2) One of the most interesting results (Figure 5b) is that individuals with the heterozygous genotype A/T at the main SNP in the MS4A7 gene are associated with significantly higher BIO16 values (strong precipitation). This indicates a possible heterozygote advantage (overdominance). This pattern is less common than directional selection and deserves deeper discussion. Can the authors propose a biological hypothesis explaining why heterozygotes might be better adapted to high precipitation conditions? Is this related to a balance between two different functions (e.g., immune response and metabolism) controlled by different alleles? In the "Discussion" section, more attention should be given to this observation, discussing possible mechanisms of overdominance and comparing them with known examples in other species.
3) In Section 2.1, it is stated that SNPs were filtered by a missing rate > 0.9. Typically, a threshold of < 0.1 or < 0.05 is used (i.e., SNPs with more than 10% or 5% missing data are removed). A value > 0.9 means that only SNPs with more than 90% missing genotypes were removed, which is a very lenient filter. Is this a typo? If not, why was such a high threshold chosen?
4) In the "Introduction" and "Discussion" sections, the authors rightly point out that precipitation is an important factor. However, the hypotheses about the specific mechanisms of this selective pressure could be more clearly formulated. The introduction could be strengthened by suggesting that high levels of precipitation are associated not only with hydric stress but also with a higher pathogen load (bacteria, parasites), the availability of certain types of feed, etc. This would make the subsequent results (enrichment of immune pathways) even more expected and logical.
5) In Section 3.1, the authors note positive Tajima's D values for all groups and interpret this as a possible result of balancing selection or demographic events such as a "bottleneck." A reduction in population size (bottleneck) usually leads to a decrease in polymorphism and can result in positive Tajima's D values. Given the history of domestication and breed formation, this demographic scenario seems very likely. It might be worth emphasizing this point more strongly.
6) In Section 2, there is no description of the methodology for conducting Redundancy Analysis (RDA). The "Results" section mentions that allele frequencies for 226,394 LD-pruned SNPs were used. This information should be included in the methods. It is stated that six variables (BIO2, BIO3, BIO4, BIO8, BIO15, BIO16) were used, but it is not described how exactly they were included in the RDA model. How was the statistical significance of the model as a whole and each of the canonical axes (RDA1, RDA2, etc.) assessed? Typically, a permutation test is used for this.
Author Response
Comment: The manuscript presents a study on the genomic adaptation of Chinese indigenous pig breeds to climatic conditions. The authors employ an approach that integrates whole-genome data, environmental variables, various genotype-environment association (GEA) methods, and multi-layered functional annotation. The study identifies precipitation (BIO16) as a key factor in selection and provides compelling evidence for the role of the MS4A7 gene in adaptation. This work is significant for understanding the mechanisms of local adaptation in livestock and can serve as a basis for developing programs for the conservation of genetic resources and the creation of climate-resilient breeds.
Overall, the manuscript is clearly written, the methodology is sound, and the conclusions are well-supported by the results. However, there are several points that require clarification or further discussion to make the article even more convincing.
Response: We sincerely thank the reviewer for the positive evaluation of our work and for recognizing the significance of our findings in understanding the mechanisms of local adaptation in livestock. We also appreciate the constructive comments provided. Following the reviewer’s suggestions, we have carefully revised the manuscript and addressed each point in detail. The revisions include clarifying methodological details, expanding the discussion on key findings such as the role of BIO16 and the MS4A7 gene, and incorporating additional analyses to improve the robustness and interpretability of our results. All modifications have been highlighted in the revised manuscript for ease of review. Point-by-point responses are provided below.
Comment 1:In Section 2.2, it is stated that "one representative latitude-longitude coordinate was selected for each breed" to reflect its long-term local adaptation. This is a critically important point, as the accuracy of GEA analysis directly depends on the accuracy of genotype-environment linkage. The ranges of indigenous breeds can be quite extensive. What was the criterion for selecting this single point? Was it the geographic center of the historical range of the breed, the location of a specific farm from which the samples were taken, or some other parameter? The methodology for selecting coordinates should be described in detail. It would also be useful to discuss in the "Discussion" section the potential limitation associated with this simplification and assess how possible inaccuracies might affect the results.
Response 1: We determined a representative latitude–longitude coordinate for each indigenous breed based on the geographic center of its historical distribution and used the long-term environmental mean at that location as the environmental phenotype. This approach is more robust than using sampling coordinates, as environmental variables act as selective pressures over long evolutionary timescales. For widely distributed breeds, we excluded commercial/hybrid populations and divided broad-range breeds (e.g., Tibetan pigs) into subpopulations by region. While some heterogeneity may remain, we believe this approach minimizes bias and ensures robustness.
Modification location: Discussion, lines 419–425 — Added detailed explanation of coordinate selection criteria and discussion of potential limitations.
Comment 2: One of the most interesting results (Figure 5b) is that individuals with the heterozygous genotype A/T at the main SNP in the MS4A7 gene are associated with significantly higher BIO16 values (strong precipitation). This indicates a possible heterozygote advantage (overdominance). This pattern is less common than directional selection and deserves deeper discussion. Can the authors propose a biological hypothesis explaining why heterozygotes might be better adapted to high precipitation conditions? Is this related to a balance between two different functions (e.g., immune response and metabolism) controlled by different alleles? In the "Discussion" section, more attention should be given to this observation, discussing possible mechanisms of overdominance and comparing them with known examples in other species.
Response 2: Breed-level genotype frequency analysis of the top SNP (2_11304356_T_A) revealed higher heterozygote prevalence in high-precipitation regions, supporting heterozygote advantage in humid, pathogen-rich environments (Supplementary Fig. S3). MS4A7 is expressed in immune- and barrier-related tissues, suggesting a role in immunoregulation under these conditions. We expanded the discussion to propose potential mechanisms of overdominance, including functional complementation of alleles and balanced immune-metabolic trade-offs, citing examples from MHC in vertebrates. We also note co-occurrence of balancing and directional selection signals, indicating a complex adaptive scenario.
Modification location: Results, lines 381–385 — Added breed-level genotype frequency analysis.
Discussion, lines 501–506 — Expanded discussion on overdominance mechanisms with cross-species examples and literature references.
Comment 3:In Section 2.1, it is stated that SNPs were filtered by a missing rate > 0.9. Typically, a threshold of < 0.1 or < 0.05 is used (i.e., SNPs with more than 10% or 5% missing data are removed). A value > 0.9 means that only SNPs with more than 90% missing genotypes were removed, which is a very lenient filter. Is this a typo? If not, why was such a high threshold chosen?
Response 3: This was a typographical error. The correct criterion is a missing rate < 0.1.
Modification location: Methods, Section 2.1, line 104 — Corrected missing rate threshold.
Comment 4:In the "Introduction" and "Discussion" sections, the authors rightly point out that precipitation is an important factor. However, the hypotheses about the specific mechanisms of this selective pressure could be more clearly formulated. The introduction could be strengthened by suggesting that high levels of precipitation are associated not only with hydric stress but also with a higher pathogen load (bacteria, parasites), the availability of certain types of feed, etc. This would make the subsequent results (enrichment of immune pathways) even more expected and logical.
Response 4: We expanded the Introduction (lines 74–86) to describe how BIO16 (precipitation of the wettest quarter) may influence adaptation via host–pathogen dynamics, thermoregulation, and hydric stress tolerance, and how increased precipitation may select for immune, mucosal barrier, and metabolic adaptations. Relevant studies in birds and domestic ruminants are cited. In the Discussion (lines 518–526), we highlight the applied significance of BIO16-associated loci in conservation and breeding.
Modification location: Introduction, lines 74–86 — Added ecological and biological background on BIO16.
Discussion, lines 518–526 — Linked BIO16 mechanisms to conservation and breeding applications.
Comment 5:In Section 3.1, the authors note positive Tajima's D values for all groups and interpret this as a possible result of balancing selection or demographic events such as a "bottleneck." A reduction in population size (bottleneck) usually leads to a decrease in polymorphism and can result in positive Tajima's D values. Given the history of domestication and breed formation, this demographic scenario seems very likely. It might be worth emphasizing this point more strongly.
Response 5:We expanded the discussion to note that positive Tajima’s D may reflect complex demographic histories such as gene flow or bottlenecks, but is also sensitive to population structure, which can inflate values. We emphasize cautious interpretation alongside other evidence.
Modification location: Discussion, lines 428–438 — Added explanation of demographic scenarios and structure-related bias in Tajima’s D.
Comment 6:In Section 2, there is no description of the methodology for conducting Redundancy Analysis (RDA). The "Results" section mentions that allele frequencies for 226,394 LD-pruned SNPs were used. This information should be included in the methods. It is stated that six variables (BIO2, BIO3, BIO4, BIO8, BIO15, BIO16) were used, but it is not described how exactly they were included in the RDA model. How was the statistical significance of the model as a whole and each of the canonical axes (RDA1, RDA2, etc.) assessed? Typically, a permutation test is used for this.
Response 6:We added detailed RDA methodology to the Materials and Methods (lines 150–153), specifying that allele frequencies for 226,394 LD-pruned SNPs and six environmental variables (BIO2, BIO3, BIO4, BIO8, BIO15, BIO16) were used. A permutation test (n = 999) confirmed all predictors were significant (P < 0.01). The first three constrained axes explained 44.5%, 18.8%, and 13.3% of the constrained variance, respectively (Supplementary Tables 6–7).
Modification location: Methods, lines 150–153 — Added RDA methodological details.
Results, lines 270–279 — Added statistical results for significance and variance explained.
Round 2
Reviewer 3 Report
Comments and Suggestions for Authors
To Authors,
The authors have adequately addressed the concerns, and I am satisfied with the revisions provided in this version.
Best Regards
Reviewer
Author Response
Comments: The authors have adequately addressed the concerns, and I am satisfied with the revisions provided in this version.
Response: We sincerely thank the reviewer for their careful evaluation and valuable comments on our manuscript. We are pleased to note the following positive feedback: “The authors have adequately addressed the concerns, and I am satisfied with the revisions provided in this version.”
We greatly appreciate the reviewer’s recognition of our work and have now submitted the revised manuscript for editorial consideration.
Thank you again for the opportunity to improve our manuscript.
Reviewer 4 Report
Comments and Suggestions for Authors
Accept in present form
Author Response
Comment: Accept in present form
Response :We sincerely thank the reviewer for their time and thoughtful evaluation of our manuscript. We are very grateful for the positive recommendation to accept the current version.
We truly appreciate your constructive comments and support throughout the review process. Your feedback has been invaluable in improving the quality of our work.
Sincerely